# Assessment of Alkali–Silica Reaction Potential in Aggregates from Iran and Australia Using Thin-Section Petrography and Expansion Testing

**DOI:** 10.3390/ma15124289

**Published:** 2022-06-17

**Authors:** Pezhman Kazemi, Mohammad Reza Nikudel, Mashalah Khamehchiyan, Paritosh Giri, Shima Taheri, Simon Martin Clark

**Affiliations:** 1Department of Geology, Faculty of Basic Sciences, Tarbiat Modares University, Tehran 14115-111, Iran; khamechm@modares.ac.ir; 2Department of Physics and Astronomy, Macquarie University, Sydney, NSW 2109, Australia; paritosh.giri@mq.edu.au; 3Department of Earth and Environmental Sciences, Macquarie University, Sydney, NSW 2109, Australia; shima.taheri@mq.edu.au; 4School of Engineering, Macquarie University, Sydney, NSW 2109, Australia; simon.clark@mq.edu.au

**Keywords:** concrete aggregate, optical thin-section petrography, SEM-EDS, XRD, expansion testing, deleterious alkali–silica reaction (ASR)

## Abstract

The alkali–silica reaction can shorten concrete life due to expansive pressure build-up caused by reaction by-products, resulting in cracking. Understanding the role of the aggregate, as the main reactive component, is essential for understanding the underlying mechanisms of the alkali–silica reaction and thereby reducing, or even preventing, any potential damage. The present study aims to investigate the role of petrographic studies along with accelerated tests in predicting and determining the potential reactivity of aggregates, including granite, rhyodacite, limestone, and dolomite, with different geological characteristics in concrete. This study was performed under accelerated conditions in accordance with the ASTM C1260 and ASTM C1293 test methods. The extent of the alkali–silica reaction was assessed using a range of microanalysis techniques including optical microscopy, scanning electron microscopy, energy-dispersive X-ray analysis, and X-ray powder diffraction. The results showed that a calcium-rich aggregate with only a small quantity of siliceous component but with a higher porosity and water adsorption rate can lead to degradation due to the alkali–silica reaction, while dolomite aggregate, which is commonly considered a reactive aggregate, showed no considerable expansion during the conducted tests. The results also showed that rhyodacite samples, due to their glassy texture, the existence of strained quartz and quartz with undulatory extinction, as well as the presence of weathering minerals, have a higher alkali-reactivity potential than granite samples.

## 1. Introduction

The construction industry is an integral part of the Iranian economy, with over two thousand active mines, the majority of which produce building and construction materials [1]. Various types of igneous and sedimentary rocks are extracted from these mines and processed to prepare raw material for construction [2]. Petrologic data have shown that rocks in Iran were formed during the evolution and formation of a complex orogenic system [2]. Thus, metamorphic, sedimentary, and igneous rocks occur in different locations throughout the country, and are used in building and construction applications. Even though aggregate specifications have a serious implication for the durability and maintenance of concrete buildings and infrastructure, the testing and specification of these aggregates prior to construction are often overlooked or not considered. The alkali–silica reaction (ASR) has been considered one of the most harmful degradation factors affecting the durability of aggregates used in concrete [3]. There have been limit studies on the alkali silica susceptibility of aggregates and their usefulness in the construction industry in Iran [4]. Iran also has varied and diverse climates in different geographical regions with relative humidity (RH) levels and temperatures varying seasonally. Previous studies have shown that above certain RH levels (>60%) and temperatures, ASR could be triggered [1,5]. Therefore, there is always a chance of ASR in concrete structures when ASR-susceptible aggregates, adequate moisture, and elevated temperatures are present [6,7].

Natural aggregates mined from igneous and sedimentary rock formations are commonly used in concrete due to their strength, wide availability, and ease of use [8]. The characteristics of natural aggregates vary depending on a range of factors, including the method of emplacement and any subsequent alteration. These determine key properties such as the chemical and structural stability of the aggregate as well as the chemical composition, in particular, the amount of silica present in the rock [9]. This can lead to greater or lesser extents of chemical reactivity between an aggregate and the cement binder when they are used to form concrete [10]. In this work, we are particularly interested in assessing the effect of rock type, texture, and mineral content on deleterious alkali–silica reactions (ASRs). The term ASR describes a reaction between susceptible minerals present in certain aggregates, engaging in a chemical reaction to form an expansive reaction product that negatively affects concrete durability [11]. This reaction occurs when aggregates containing active forms of SiO_2_ react with alkalis in the pore water to produce crypto-crystalline to amorphous alkali–silica gels. These gels increase in volume when they absorb water, creating a build-up of internal pressure and giving rise to subsequent cracking of the concrete and loss of structural strength [12,13]. The reactivity of an aggregate depends on various factors, including geological origin, mineral constitution, and texture [14,15,16]. Reactive aggregates contain highly reactive forms of silica (opal, cristobalite, tridymite, acid volcanic glass) which have a high tendency to react with alkalis (potassium and sodium) [14,16]. It is possible for aggregates from the same origin or even the same bedrock to have different ASR potentials [17,18,19]. The varying ASR susceptibility of aggregates is highly dependent on their fabric and mineralogy [20]. The most effective way to recognize reactive aggregates is through optical thin-section petrography, and the verification of inconclusive results through expansion testing [21,22]. Being able to predict and detect deleterious aggregates could reduce the chance of cracking and could extend the service life of a concrete structure. It is widely accepted that three components have to be present for structural damage caused by ASR, including high alkali content in the cement paste, the presence of reactive forms of silica in aggregates, and the presence of an adequate amount of moisture [23,24]. Highly reactive aggregates commonly contain very fine-grained quartz and amorphous forms of silica (for example, opal and chalcedony), while the more slowly reactive aggregates typically contain crystalline quartz-bearing rocks (mylonite, granite, gneiss, quartzite, greywacke, phyllite, and argillite) [25].

The petrographic method described in RILEM AAR-1.1 [26] and AAR 1.2 [27] is considered the first step in the evaluation of the potential alkali reactivity of virgin aggregate materials. This method is generally used to identify rock types and minerals that might react with hydroxyl ions from the concrete pore solution. Analytical techniques such as scanning electron microscopy with energy-dispersive spectroscopy (SEM/EDS) and X-ray diffraction (XRD) are widely used for the detailed identification of by-products of the ASR process and are considered a complementary physicochemical method to the petrographic approach [28,29,30]. The petrographic method is followed by accelerated laboratory tests based on RILEM AAR-2 [31] and AAR-3 [32]. In these accelerated evaluation approaches, mortar or concrete bars are exposed to severe conditions of alkalinity and temperature to initiate expansion within days, weeks, or years, depending on the method. Thus, the most popular and effective methods of detecting the ASR potential of aggregates are the accelerated mortar bar test (AMBT) based on ASTM C1260 [33] and CPT according to ASTM C1293 [34].

This study is, therefore, designed to investigate the susceptibility of commonly used aggregates to ASR. Two igneous and three sedimentary aggregates from Iran and Australia were chosen for this purpose. AMBT and CPT were performed based on ASTM C1260 [33] and ASTM C1293 [34], respectively, to assess the expansion of samples. Microstructural analysis and the diffraction profile were evaluated using SEM/EDS and X-ray powder diffraction (XRPD) to obtain an in-depth understanding of the ASR susceptibility of the tested aggregates. 

## 2. Materials and Methods

### 2.1. Sample Materials

In the present study, two silica-rich igneous rocks, including rhyodacite and granite, and three sedimentary carbonate rocks, including Bathurst limestone, Ilam limestone, and dolomite, locally used as concrete aggregate, were characterized in detail. The granite sample was collected from Nehbandan County in Iran’s South Khorasan Province. The rhyodacite sample was collected from Yazd Province in Iran. The dolomite sample was collected from Damavand City in the Tehran Province of Iran. The Bathurst limestone sample and Ilam limestone were collected from Bathurst, New South Wales, in Australia and from Dehdasht County in Kohgiluyeh and the Boyer-Ahmad province of Iran, respectively.

The mineral modal content of the sample materials was characterized using optical microscopy equipped with a digital camera as well as through XRPD and X-ray fluorescence spectrometry (XRF) methods. The sulphate-resisting cement was obtained from Boral Cement Australia, and it complies with the Australian Standard AS3972-2010 [35] as type-SR cement.

### 2.2. Physical Properties

The physical and mechanical properties of the samples in the virgin state were evaluated using the methods suggested by ISRM [36] from the prepared cylindrical cores. The mean saturated (γ_sat_) and dry unit weights (γ_d_), effective porosity (n_e_), and water absorption (W_a_) were calculated using the saturation and buoyancy technique [36]. The physical properties were measured for three samples and averaged. In accordance with the ISRM-suggested method [36], S-wave and P-wave velocities were evaluated in a dry state using a Pundit Lab ultrasonic test device. Three uniaxial compressive strength tests were conducted for each rock. 

### 2.3. Expansion Testing through C1260-07 (AMBT)

Mortar bars (25 × 25 × 285 mm) were prepared by mixing aggregate (0.125–5.00 mm fraction) with SR cement and water in a 2.25:1.00:0.47 ratio (aggregate:binder:water). For each aggregate type, four mortar bars were prepared using a single mix batch (ASTM C1260-07).

The accelerated mortar bar test (AMBT) was conducted according to the ASTM C1260-07 standard to measure the expansion of mortar bars 25 × 25 × 285 mm. The AMBT is designed for the rapid assessment of non-reactive aggregates (<0.1% expansion of AMBs) and reactive aggregates (>0.1% expansion of AMBs). After 24 h hardening and 24 h curing of mortar bars, the initial measurement of the length was taken. The mortar bars were then immersed in a 1M NaOH solution and placed in an oven at 80 °C. The length of the mortar bars was measured at regular intervals for 28 days to check the expansion. The casting of the mortars and length comparator are shown in Figure 1.

### 2.4. Expansion Testing through C1293-08 (CPT)

Concrete prisms (75 × 75 × 300 mm) were prepared according to ASTM C1293-08 to assess the long-term expansion of samples. The CPT is used for the long-run assessment of the reactivity of aggregates; if the expansion of concrete prisms exceeds 0.04%, they are considered reactive aggregates, while for expansion below 0.04%, they are classified as non-reactive aggregates. After 24 h hardening and 24 h curing of the concrete prisms, the initial measurement of the length was taken. The concrete prisms were then immersed in a 1N NaOH solution and placed in an oven at 38 °C. The length of the concrete prisms was monitored over a period of one year based on a specific timetable to examine their expansions. Figure 2 shows the prepared concrete prism samples and length comparator.

### 2.5. Thin-Section Preparation and Optical Petrography

Standard-sized ~27 × 47 mm thin sections of a 30 µm nominal thickness were prepared by cutting billets (~25 × 25 × 45 mm), following routine procedures at Macquarie University facilities. The billets were mounted on carrier glass using Struers’ EpoFix epoxy and lapped to thickness with silicon carbide slurries. The surface was finished with diamond pastes 6-3-1 µm on cloth to obtain a mirror shine. Sections were prepared from virgin aggregate particles as well as from mortar bars and concrete prims post-mortem after expansion testing was completed (Figure 3).

The finalized sections were studied under a Bell MPL-2 microscope equipped with an LED fluorescence lamp. A series of photographs was taken from the entire surface of the thin section with a 3-megapixel CC fluorescent camera in polarized light and fluorescent light. The micrographs were developed with a Microsoft Image Composite Editor (ICE) to obtain an overview of microstructure and texture/fabric.

The mineral modal contents in vol% were estimated using the nomograms in Terry and Chilingar [37]. Rock types were classified according to the current IUGS-approved nomenclature. In figures and tables, the mineral names are abbreviated using the acronyms of Whitney and Evans [38].

### 2.6. Scanning Electron Microscopy (SEM)

To minimize charging under the impinging electron beam, polished section surfaces were coated with ~20 nm of carbon in a BOC Edwards 306 Auto vacuum coater through thermal evaporation. The samples were prepared from selected mortar bars exposed to elevated conditions per ASTM C1260-07 for 1, 5, 9, or 14 days, or from concrete prisms exposed for 1, 3, or 6 months to ASTM C1293-08 conditions, to verify the presence of alkali–silica reaction products.

The coated sections were studied in a Zeiss EVO MA15 SEM instrument, operated in high vacuum (<5 × 10^−6^ Torr) at a 15 kV acceleration voltage and a 500 pA beam current (on Faraday cup). The phases were identified in situ through chemistry using an Oxford Instruments X-MAX (Oxford, UK) 20 mm^2^ energy-dispersive spectrometry silicon drift detector EDS-SDD at a 2.0 nA beam current, a dwell time of 30 s, and a detector dead time <20%. Raw element data were ZAF corrected using the PAP algorithm of Pouchou and Pichoir [39], as implemented in the AZtec proprietary software. The corrected element contents were converted to oxide contents in weight percent (wt%) assuming stoichiometry; Fe_2_O_3_^T^ represents total iron (oxide) content. EDS analysis was used uncalibrated for phase identification purposes only; no quantitative data were acquired.

### 2.7. Mineral Content through XRPD Analysis

The XRPD analysis was conducted on the aggregates and the samples exposed to ASTM C1260-07 conditions for 1, 14, and 21 days. The XRPD method was used to identify the phase of crystalline materials in these aggregates and the formation of any reaction products with time. The method was also used as a supplementary measure to identify secondary alteration of cement hydrates.

The pieces of mortar bars and the fresh aggregates were finely powdered separately using the mortar and pestle. The small portion of powdered samples was further pulverized to obtain an evenly distributed ~50 μm grain size. The grinding duration was kept constant for all samples to achieve the highest possible reproducibility and repeatability. The fine powder was mixed with silicon powder and packed into a sample holder. The surface of the packed powder was pressed and smoothed with a piece of float glass.

Mounted specimens were analysed in a PANalitical X’Pert Pro diffractometer instrument with a Cu tube operated at 45 kV and 40 mA. Diffractograms were recorded using unfiltered bulk Cu*K*α radiation at λ = 1.54184 Å, from 05–90 °2θ at 0.01 °2θ increments, with 24 s counting time per step, overall scan time 56H40M. Ilam limestone was only scanned over 05–60 °2θ. Intensities are plotted in arbitrary units (a.u.) against diffraction angles in °2θ.

## 3. Results

### 3.1. Optical Thin-Section Petrography

Petrographic examinations with the use of an optical microscope equipped with a digital camera along with XRPD and XRF results demonstrated more information about the compositions, characteristics, and microstructures of the different aggregates and their minerals. Figure 4 shows the microstructural and mineral compositions of the five mentioned aggregates. The XRPD and XRF results are shown in Table 1 and Table 2.

#### 3.1.1. Granite

In the thin section, the Nehbandan granite had a granular fabric which consisted of quartz (Qz) (30 vol%), K-feldspar (Kfs) (20 vol%), plagioclase (Pl) (30 vol%), Biotite (Bt) (10 vol%), and amphibole (Amp) (≈5 vol%) as the main mineral components, along with a small amount of minerals such as iron oxide and calcite (Figure 4a). The microscopic examinations also revealed the sub-graining and presence of microcracks in some quartz minerals, which might facilitate the infiltration of concrete pore solution, thus increasing the ASR potential of this aggregate. The Kfs grain in the top left corner in Figure 4a is clearly cloudy, which is typical for kaolinitization and/or sericitization.

#### 3.1.2. Rhyodacite

The material studied here had a porphyritic texture of plagioclase (Pl) (55 vol%), quartz (Qz) (25 vol%), and biotite (Bt) (15 vol%) phenocrysts floating in a microcrystalline matrix. Sanidine (Sa) constituted less than 5 vol% of the rock (Figure 4b). Clinoamphibole grains (Cam) typically have opaque reaction rims, occasionally replacing most of the phenocryst. Quartz sometimes displays undulatory extinction of tectonic origin [40,41,42]. Quartz sub-graining is commonly observed, as is the alteration of K-feldspar to fine-grained muscovite (sericite) plus quartz.

#### 3.1.3. Bathurst Limestone

The thin-section examinations of Bathurst limestone showed that the aggregate consisted mainly of ooid (0.25–2 mm diameter), ancoide, and aragonite intraclasts which are bonded by a calcitic cement (Figure 4c). According to the current nomenclature of Hallsworth and Knox [43], this rock type is classified as grainstone.

#### 3.1.4. Ilam Limestone

The microscopic thin-section image of Ilam limestone is shown in Figure 4d. Different kinds of fossils were observed in the thin section, including foraminifera, nomoliths, gastropods, and ammonites, cemented by a clay matrix. This rock is called packstone, according to the classification of Hallsworth and Knox [43].

#### 3.1.5. Dolomite

The dolomite rock consisted of compacted ooid packstone with micritic calcite cement. Fine dolomite crystals (amorphous and semi-shaped) were observed, along with a micrite matrix in the examined thin sections. Moreover, oxide and sulphide minerals can be seen along the broken and fractured surfaces (Figure 4e). Figure 4f shows an optical microscope equipped with a digital camera and a polarized light source, which were used for petrographic examinations.

### 3.2. Physical and Mechanical Properties of Aggregates

The physical and mechanical characteristics of the aggregates are shown in Table 3. According to the table, rhyodacite and Bathurst limestone had the highest effective porosity (n_e_) at about 3.65% and 2.22%, and water absorptions (W_a_) of 1.42% and 1.45%, respectively. The porosity and water absorption capacity of aggregates are considered effective factors in the dissolution of silica minerals by alkali ions [44,45,46,47,48,49]. On the other hand, granite aggregate had the lowest effective porosity and water absorption, about 0.7% and 0.27%, respectively. In addition, the Bathurst limestone aggregate had the lowest compressive strength (UCS), which was 48.78 Mpa, while the granite sample has the highest compressive strength, around 112.6 MPa.

### 3.3. Mortar Bar Expansion

The changes in the expansion of the mortar bars with increasing immersion duration for all the samples are shown in Figure 5. According to ASTM C1260 limits [33], an aggregate is classified as reactive if expansion exceeds 0.1% in 14 days [50]. The guideline specified by ASTM C1260 is shown as a solid grey block, where the expansion below the horizontal line (0.1%) indicates non-reactive aggregates, and above the line, reactive aggregates [33]. Based on this test, Bathurst limestone and rhyodacite aggregates exceeded the limit of 0.1% after 14 days of testing, while granite, Ilam limestone, and dolomite showed a lower value. Therefore, Bathurst limestone and rhyodacite can be classified as reactive, while granite, Ilam limestone, and dolomite can be regarded as non-reactive. However, it should be noted that the expansion in rhyodacite exceeded the limit by only 0.01%.

### 3.4. Post-Mortem Mortar Bar Microstructure through SEM 

Microstructural observations were performed on polished surfaces of sliced samples using SEM. In Figure 6 and Figure 7, the comparison of the SEM micrographs of the aggregate samples after 1-day and 14-day immersion in 1N NaOH solution at 80 °C is shown. Figure 6a,b show 1- and 14-day samples made of granite grains. Fourteen days after the start of the test, it was observed that there was no considerable evidence of reactivity, such as expansion cracks in the aggregates or gel formation at the boundary between the cement paste and aggregates or inside the aggregates, which indicates that the granite aggregates do not react with alkali ions, based on the mortar bar test results and SEM analysis. Due to the fact that the granite is composed mostly of quartz minerals which contain a crystalline structure, as well as the lack of tectonic tension effects in the examined thin section, such as undulous extinction, considered a sign of the reactivity of quartz minerals, the low expansion of the granite aggregate is reasonable [41,42].

Figure 6c shows that expansion cracks were formed in the boundary between cement paste and rhyodacite aggregates. According to the elements analysis, it can be concluded that these cracks were formed due to the formation of alkaline gel in the boundary between the aggregates and cement paste. The main elements composing ASR reaction product include silicon, potassium, sodium, and calcium, with minor amounts of aluminium, magnesium, and iron [41,42,51,52,53]. The typical ratio of (Na + k)/Si and Ca/Si is about 0.2–0.4 [30,53,54,55,56,57]. As can be seen from Figure 6d, in the elemental analysis of the marked point in the rhyodacite 14-day samples, the (Na + k)/Si and Ca/Si ratios were 0.24 and 0.32, respectively, which is relatively similar to the composition of typical alkali gels; therefore, it was concluded that these products are related to the formation of alkali–silica gel. In the petrographic examinations, it was determined that the rhyodacite samples had a glassy-to-microcrystalline matrix and silica content, and we detected weathering products such as sericite and secondary quartz together with the presence of undulous extinction in some quartz crystals, leading to the idea that rhyodacite aggregates have a relatively high potential for alkali reactivity.

Figure 7a,b show the 1- and 14-day micrographs of Bathurst limestone. The evidence of expansion occurring inside the Bathurst limestone aggregate, as well as at the interface between aggregates and the cement paste after 14 days of storing the samples under an accelerated condition, is shown in Figure 7b. Micro-cracks were observed along the aggregates and cement paste interface area, as well as grain boundaries and microspores. Furthermore, SEM images of Bathurst limestone samples contain numerous cracks within the aggregates, as demonstrated by the highlighted section in Figure 7b. ASR can be verified based on networks of micro-cracks and the presence of alkali–silica gels (ASGs).

Figure 7c,d show electron microscope images of 1- and 14-day samples of mortar made with Ilam limestone aggregate. After 14 days of the testing process, no significant changes were observed in terms of the expansion cracks or the presence of alkali gel inside the aggregates, nor in the transition zone between the aggregates and the cement paste.

Although, with the passage of time, new cracks were observed in the dolomite aggregates, due mainly to the dedolomitization process, there was no sign of alkali gel in the observed cracks (Figure 7e,f). In the dedolomitization process, magnesium ions are replaced by calcium; as a result, due to the lower atomic radius of magnesium ions, the number of cavities in these samples increases, which can cause cracking in these aggregates [50,58,59,60,61].

### 3.5. Paste Development through XRPD

The diffraction profile was indexed using XRPD. The XRPD results showed that peak intensity generally increased with time, due mainly to the reaction between aggregates and cement and an increase in the crystallization of minerals. There were a few peaks which decreased in intensity with time. This can be attributed to the mineral’s growth or shrinkage in the concrete [62]. The appearance of new peaks was observed, due to the reaction between the existing minerals producing new reaction products with time. The XRPD intensity plots of 1-day accelerated mortar samples were offset by 2000 and the plot of 21-day accelerated mortar samples by 4000 to clearly show the change in intensity between the peaks.

The XRPD intensity comparison between the granite aggregate accelerated mortar for 1 day and 21 days is shown in Figure 8a. There were multiple new peak formations at 18°, 29°, 34°, and 47° in the accelerated sample. The peaks at 18°, 34°, and 47° are due to the portlandite formation, while the peak at 29° is probably due to the formation of albite.

The XRPD intensity comparison between 1-day and 21-day rhyodacite aggregate mortars is shown in Figure 8b. There was a new peak formation at 18° in the accelerated samples, due mainly to the portlandite formation. There was no indication of ASR from the XRPD intensity comparison in rhyodacite aggregate. The XRPD intensity pattern of 1-day and 21-day mortars of Bathurst limestone aggregate is shown in Figure 8c. A new peak formation is seen at 18° and 34° in the 21-day accelerated sample, compared to the 1-day mortar. These peaks matched the XRPD pattern of portlandite. However, there was no sign of any reaction product forming due to the ASR. The ASR in limestone occurs mainly as a result of the expansion of clay minerals (which provides SiO_2_ for the reaction) in the aggregate matrix [24,63,64].

Figure 8d shows the XRPD intensity comparison between 1 day and 21 days of Ilam limestone aggregate accelerated mortar. There was some peak growth at 18 °2θ and 34 °2θ of the 21-day mortars compared to the 1-day sample, which is related to the formation of portlandite and calcite. The silica peak was detected at 28 °2θ of the 1-day and 21-day mortar samples. However, there was no sign of formation of new peaks with the passage of time.

Figure 8e shows the XRPD intensity comparison of 1-day and 21-day dolomite mortar samples. There were new peak formations at 18°, 26°, 47°, and 57° in the accelerated sample. The peaks at 18° and 47° matched the XRPD pattern of portlandite, formed due to the reaction between elemental calcium and water during the curing process [65,66]. The peak at 26° matched the XRPD pattern of silica. Dolomite dissolves into Ca^2+^, Mg^2+^, and ${\mathrm{CO}}_{3}^{2-}$ ions and combines with water molecules under the attack of alkaline solutions, which resulted in calcite crystallization. The calcite is formed as a product of dedolomitization when portlandite consumes ${\mathrm{CO}}_{3}^{2-}$ ions [24,67,68].

Although dolomite, Bathurst limestone, and Ilam limestone contain silica, the XRPD intensity of the dolomite and Ilam limestone showed crystalline silica which was not present in Bathurst limestone. This suggests the presence of amorphous silica in Bathurst limestone, which is the chief cause of ASR in the calcareous aggregates [24,53,69].

The detailed analysis of different XRPD intensity peaks at different ages demonstrated the possibility of ASR reaction in Bathurst limestone, but there was no clear evidence of the reactivity of other aggregates. Although AMBT, CPT, optical microscopy, and SEM results showed the alkali reactivity potential of Bathurst limestone and rhyodacite, the XRPD results of powder diffraction did not show the reactions in these aggregates.

### 3.6. Concrete Prism Expansion

The changes in the expansion of the concrete prism with increasing immersion duration for all the samples are shown in Figure 9. According to the ASTM C1293 standard [34], an aggregate is classified as reactive if expansion exceeds 0.04% in 365 days. Based on this test, rhyodacite and Bathurst limestone aggregates exceeded the limit of 0.04% after 120 and 150 days, respectively, while granite, Ilam limestone, and dolomite showed lower values. Therefore, Bathurst limestone and rhyodacite can be classified as reactive aggregates, while granite, Ilam limestone, and dolomite are non-reactive aggregates.

### 3.7. Post-Mortem Concrete Prism Microstructure

Microstructural and elemental examinations were conducted on polished surfaces and thin-section samples taken from concrete prisms using backscatter scanning electron microscopy (BSEM). The SEM micrographs of the different mortar prisms after 1-, 30-, 90-, and 180-day immersion in 1M NaOH solution at 38 °C were taken and analysed. Micro-cracks were observed along the aggregate and cement paste interface area, as well as grain boundaries and micropores. As mentioned in the accelerated mortar bar test, ASR can be verified based on networks of micro-cracks and the presence of alkali–silica gels (ASGs) [50,70,71].

Figure 10 shows BSEM images of trigonal crystals of ettringite in 6-month-old samples of Bathurst limestone. Ettringite is a hydrous calcium aluminium sulphate mineral with the formula Ca_6_;Al_2_; (SO_4_)_3_(OH)_12_ 26 H_2_O. It is a colourless-to-yellow mineral crystallizing in the trigonal system [72,73,74]. Because the images were taken from a thin section, the morphology of the ettringite is not very clear, but based on their prismatic structure, it can be concluded that they are related to alkali–silica reaction products. In the examinations performed on prepared thin sections from Bathurst limestone samples, no sign of alkaline gel was found, but according to the results of expansion tests and petrographic studies, it was determined that alkali reaction products were formed and expansion occurred in this aggregate. The ACR products of brucite and calcite were detected, but no ASR gel was found in the cracks of the tested samples.

Figure 11 shows the micrographs of dolomite concrete samples after 6 months of testing according to the ASTM C1293 standard under different magnifications. Figure 11a shows the micrograph with 10 µm magnification, while Figure 11b presents 2 µm magnification, which provided detailed observation of the cracks.

In dolomite specimens, cracking of the aggregate and concrete can occur due to the dedolomitization process [59,68,75,76]. In the tested concrete samples, the presence of cracks and fissures in the concrete aggregates can be a sign of expansion and the aggregates’ reactivity, but in a careful assessment of the cracks created in dolomitic aggregates, as well as in the transition zone between the aggregates and the cement paste, no evidence of alkali gel was observed. In the alkali–carbonate reaction (ACR) involving dolomite, no ASG can be found; however, it may occur when the dolomitic aggregate contains microcrystalline quartz [76]. However, according to the results obtained from petrography studies as well as expansion tests, the absence of alkaline reaction products in dolomite concrete samples was not unexpected, possibly due to the very small amount of silica in the dolomite aggregate [24,77]. The absence of gel-like materials in aggregates and the low expansion value measured through AMBT and optical microscopy examinations led us to conclude that the examined dolomite aggregates have no significant potential for alkali aggregate reactivity.

### 3.8. Optical Petrography of Concrete Prisms

Thin sections of concrete prisms were prepared for microscopic studies at different times (1, 3, and 6 months) to examine their microstructural features over time. Figure 12a,b show images of 1- and 6-month granite concrete prism samples. According to the figure, no significant changes were observed in terms of crack openings and the presence of gel-like material inside the cracks over time, which confirms the high durability of this aggregate against the attack of alkaline ions such as sodium and potassium. This aligns well with the results obtained from the accelerated mortar bars and concrete prism tests, as well as the results of the SEM analysis. Figure 12c,d represent 6-month samples of rhyodacite taken in non-polarized light and in polarized light, respectively. As shown in the figure, traces of gel-like materials were observed in the samples. Alkaline silica gel turns dark in polarized light, while in non-polarized light it appears white [78,79]. As a result, it was concluded that the identified vein (indicated by the red arrow) can be related to alkaline silica reaction products.

Figure 13a,b show thin sections of 1-month and 6-month samples of Ilam limestone. As shown in Figure 13b, 6 months after the start of the experiment, no significant change in terms of cracking or in the destruction and decomposition of calcareous aggregates was observed in the thin sections, which indicates the high resistance of Ilam limestone aggregate to the alkali reaction. Microscopic examinations of these samples confirm the results obtained from the SEM examinations, accelerated mortar bars, and concrete prism tests.

Figure 13c,d display optical microscopy images of the dolomitic specimen after 1 month and 6 months. In these samples, the number of cracks and fractures increased over the time period, but no evidence of alkaline gel was observed in the cracks; it was therefore concluded that these cracks are due to the dedolomitization reaction of dolomite crystals. The results of this experiment aligned well with the results of the SEM and expansion tests.

## 4. Discussion

The present study aimed to assess the susceptibility of five different aggregates to ASR based on petrographic analysis, as well as chemical and mechanical examination of these aggregates. It has been shown that quartz from shear zones is indeed susceptible to deleterious ASR [80,81,82]. Volcanic rocks such as rhyodacite and rhyolite, due to their formation with rapid cooling rates, often contain microcrystalline or glassy matrix which are susceptible to dissolution by an alkaline environment and eventually cause ASR and cracking of concrete. Some kinds of silica with a poor crystalline structure, such as opal, chalcedony, cristobalite, and tridymite, have a greater tendency to react with the alkaline pore solution of concrete, causing deleterious ASR products [7,17,77,83,84]. In addition to silica, alkaline ions in volcanic glasses can be released into the pore solution of concrete and facilitate the ASR process [53,84]. When minerals such as micas, feldspars, clays, and zeolites are weathered, the release of alkalis is more intense and, as a result, the ASR process will be more deleterious [85,86,87]. The major contents of ASR products are silicon, sodium, potassium, and calcium, and the minor minerals are often aluminium, magnesium, and iron [51,52,53]. The typical ratio of (Na+k)/Si and Ca/Si is about 0.2–0.4 [30,53,54,55,56,57].

In the present study, the petrographic examinations showed that granite aggregates have a granular texture, including quartz (Qz), k-feldspar (Kf), plagioclases (Pl), and Biotite (Bt) grains. It should also be noted that there is no significant effect of weathering or alteration processes on the granite minerals, but rhyodacite specimens have a glassy to microcrystalline matrix, in which many of the quartz grains in the matrix are amorphous or microcrystalline, and therefore more likely to react with cement alkalis to form an alkaline silica gel. Based on the petrographic studies of rhyodacite samples, secondary minerals such as sericite and chlorite were formed as a result of the weathering and alteration of feldspar and mica minerals. Therefore, it can be concluded that the alkali reactivity of rhyodacite aggregates can be affected by the rate of weathering and alteration of primary minerals and the presence of secondary minerals. Furthermore, the petrographic examination revealed that some of the quartz minerals in the rhyodacite exhibit undulatory extinction. The existence of undulose extinction and strained quartz can be the result of tectonic or magmatism forces which make these minerals more unstable in deleterious environments and susceptible to the ASR process [88,89,90,91]. There is a relationship between the deformation of minerals and their reactivity [80,81]: the greater the deformation of the quartz grains, the greater their tendency to react with alkalis [80,81,82,90,92]. As the studied rhyodacite aggregate contains deformed quartz minerals with undulose extinction, their tendency to react with alkali ions is expected. Indeed, SEM analyses demonstrated that in the rhyodacite sample, the ratio of Ca/Si and (K + Na)/Si is 0.32 and 0.24, respectively, which is similar to those for common ASR products reported by the literature [30,53,54,55,56,57].

The results of the expansion tests (AMBT and CPT) of the rhyodacite and granite samples accorded well with the petrographic results and showed that the expansion of rhyodacite aggregate exceeded the threshold in both accelerated and long-term expansion tests, while the expansion of granite samples was lower than the threshold.

The presence of fine silica in carbonate aggregates has proven to be the major cause of deleterious ASR in many cases [24,53,93,94,95]. However, in previous studies, the expansion of carbonate aggregates was attributed to dedolomitization [96,97,98]. Although sedimentary aggregates have a small amount of silica, research studies have shown that they can undergo ASR, which is due mainly to the presence of a small-to-appreciable amount of reactive silica [24,99]. In some non-siliceous carbonate rocks, even with low silica content, the alkali–silica reaction can occur [24,94], which was observed in the tested Bathurst limestone aggregate. In the dedolomitization process, dolomite reacts with dissolved sodium and potassium in an alkaline pore solution, creating brucite magnesium hydroxide within the dolomite. Sodium, potassium, and carbonate remain dissolved and react with portlandite, making calcite calcium carbonate and forming holes of calcite in the cement paste peripheral. The process of dedolomitization in some cases can negatively affect the soundness of the aggregate and concrete, but it does not necessarily cause expansion [75,100]. The amount of reactive siliceous component in the examined dolomite was low for any ASR to occur. The innocuous nature of dolomite has been reported in multiple studies [7,95]. Islam and Akhtar have concluded that pure dolomite rocks can be considered non-reactive, while dolomitic limestone is prone to ASR [101]. Similar findings were reported by Deng et al., where they compared dolomitic limestone with argillaceous dolostone and pure dolostone and determined that dolomitic limestone is highly reactive, showing higher expansion compared to the dolostone [68].

The effect of the porosity of aggregate and concrete on the alkali reaction has been reported in the literature. Bulteel et al. reported that the high porosity of concrete facilitates the flow of moisture and alkaline ions into the concrete and aggregate, and therefore makes them more susceptible to ASR [48]. Haha et al. found that aggregates with high porosity are more susceptible to ASR [49]. In similar results, Stefan et al. [46] and Shah and Ahmed [47] showed that the resistance of concrete aggregates to ASR increases when the porosity is low.

In the present study, based on the XRF results, among the three examined carbonate samples, Bathurst limestone had a higher silica content: 4.8% compared to 1.5% in Ilam limestone and 0.15% in dolomite. Moreover, physical and mechanical examinations of the aggregates revealed that the effective porosity of Bathurst limestone (2.22%) was higher than that of Ilam limestone (1.97%) and dolomite samples (1.04%). Furthermore, Bathurst limestone had a higher water absorption (1.45%), compared with 0.84% and 0.78% in Ilam limestone and dolomite, respectively. In addition, the presence of ooid structures with calcite or silica cores, as well as the presence of aragonite intraclasts in Bathurst aggregate, can be effective in their higher porosity and reactivity. According to the findings noted above, as well as the results of the AMBT, CPT, SEM/EDS, and petrographic examinations, it was concluded that Bathurst limestone can be classified as an alkali-reactive aggregate, while Ilam limestone and dolomite were regarded as non-reactive aggregates.

## 5. Conclusions

This study investigated the reactivity of five different aggregates in cement mortar and concrete. Several techniques were employed to predict and determine the reactivity of these aggregates, including optical microscope, AMBT, CPT, SEM/EDS, BSEM, XRF, and XRPD. The main conclusions are as follows:The petrographic examinations and mechanical properties revealed that among the three studied carbonate aggregates, Bathurst limestone had the highest amount of silica content, effective porosity, and water absorption. According to these findings, it was concluded that Bathurst limestone is more susceptible to ASR.Based on the petrographic examinations, rhyodacite contains a microcrystalline-to-glassy matrix, quartz undulatory extinction, and secondary minerals such as sericite, suggesting that rhyodacite aggregate is more likely to be alkali-reactive.The reactivity of all the selected aggregates was further verified through AMBT, CPT, SEM/EDS, BSM, and optical microscopy analysis. The results confirmed that among all studied aggregates, the Bathurst limestone and rhyodacite aggregates can be classified as potentially reactive.According to the above-mentioned analyses, it was suggested that carbonate aggregates containing higher amounts of reactive silica, effective porosity, and water absorption are more susceptible to ASR. Moreover, the results showed that igneous aggregates with glassy matrix, reactive quartz, undulatory extinction, and secondary minerals, such as sericite and clay minerals, have more potential alkali reactivity. Future long-term studies are needed to identify all changes developing in concretes, as it may take a long time for some cementitious materials to release the alkali. The ASR-susceptibility evaluation of aggregates is recommended for testing the suitability of aggregates for different construction applications in locations with higher humidity levels (e.g., the Persian Gulf or near the Caspian Sea) and elevated temperatures in future work.

## Figures and Tables

**Figure 1 materials-15-04289-f001:**
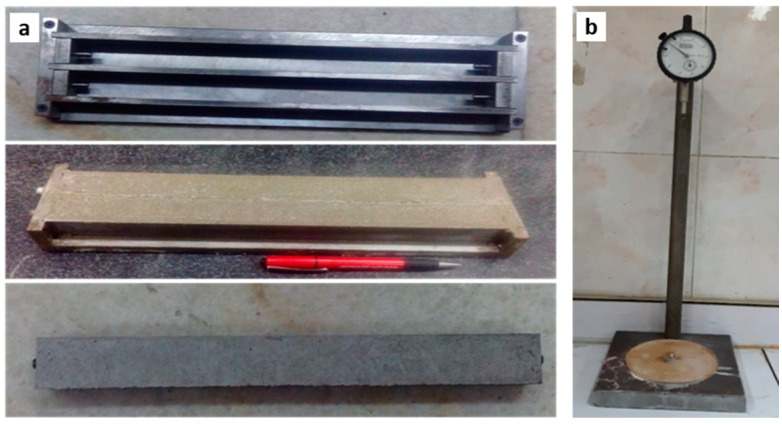
Accelerated mortar bar samples (ASTM C 1260): (**a**) casting equipment and mortar bars; (**b**) length measurement facility.

**Figure 2 materials-15-04289-f002:**
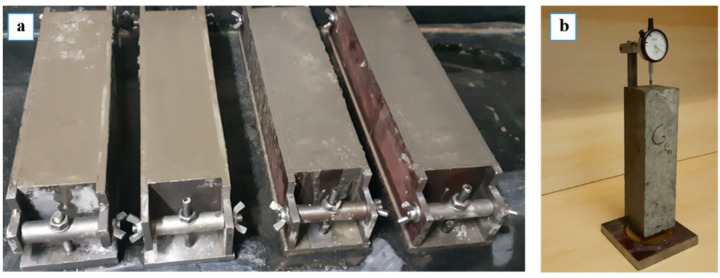
Concrete prism samples: (**a**) casting; (**b**) length measuring facility.

**Figure 3 materials-15-04289-f003:**
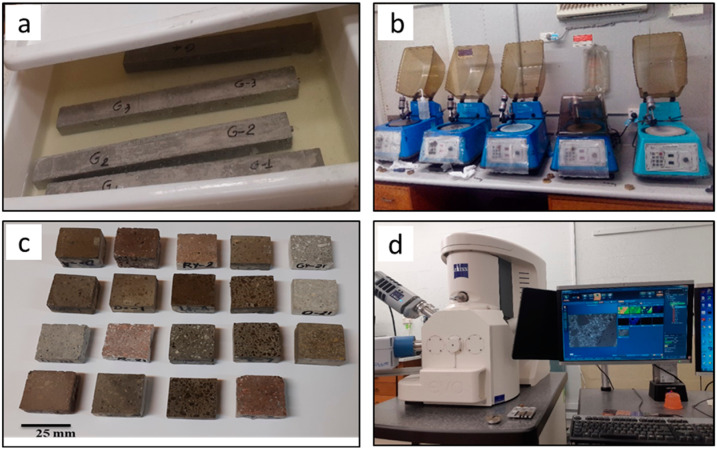
Preparing sample for SEM analysis: (**a**) mortar bars; (**b**) polishing equipment; (**c**) polished samples; (**d**) SEM facility.

**Figure 4 materials-15-04289-f004:**
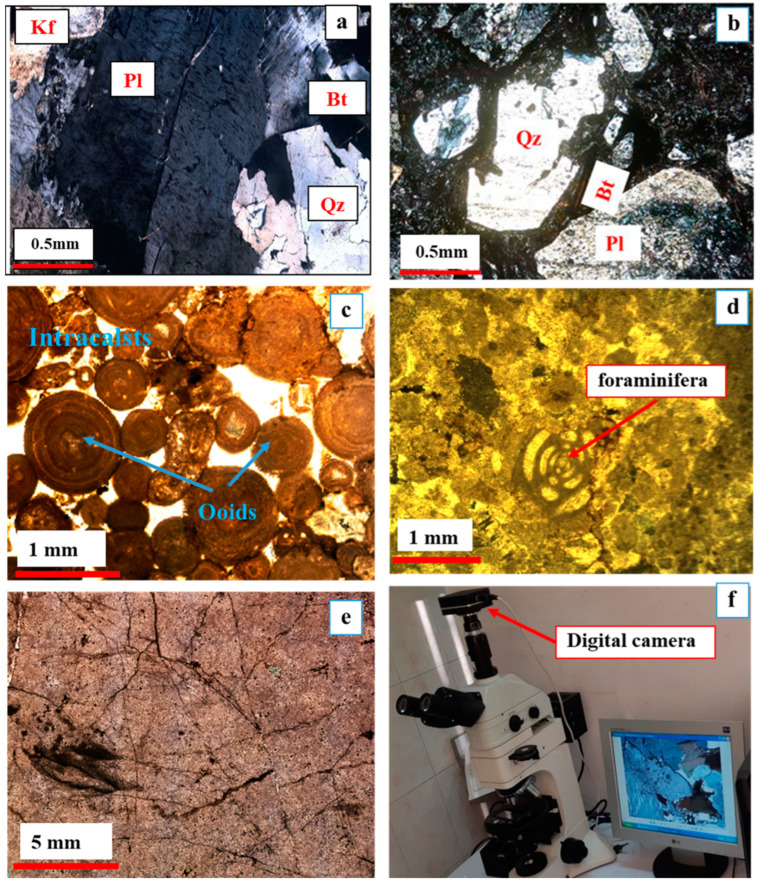
Microstructure of aggregates: (**a**) granite; (**b**) rhyodacite; (**c**) Bathurst limestone; (**d**) Ilam limestone; (**e**) dolomite; (**f**) optical microscope equipped with a digital camera and polarized light.

**Figure 5 materials-15-04289-f005:**
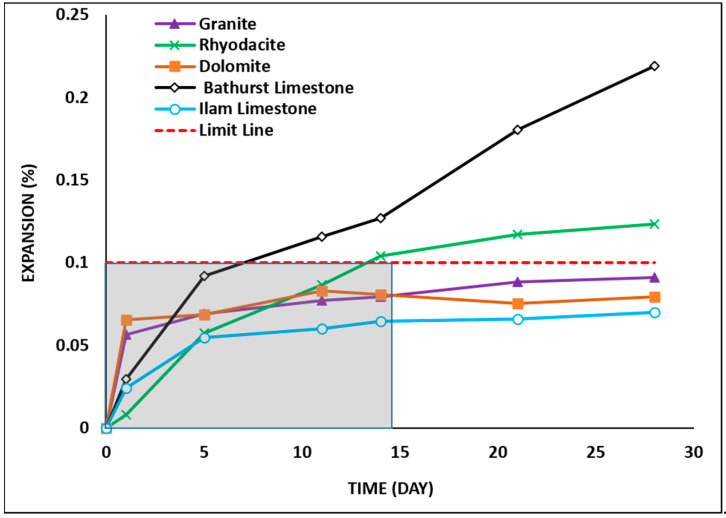
Accelerated mortar bar test results (ASTM C 1260).

**Figure 6 materials-15-04289-f006:**
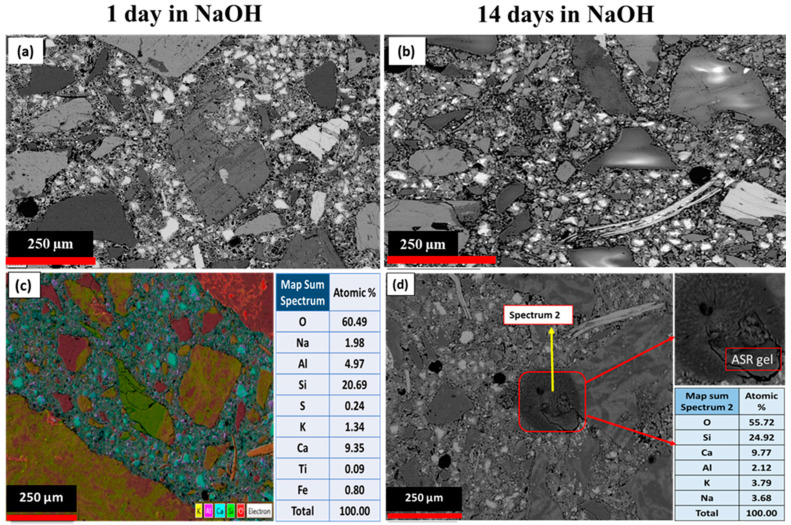
Comparison of SEM micrographs of (**a**) granite (1 day), (**b**) granite (14 days), (**c**) rhyodacite (1 days) and (**d**) rhyodacite (14 days) immersed in NaOH solution.

**Figure 7 materials-15-04289-f007:**
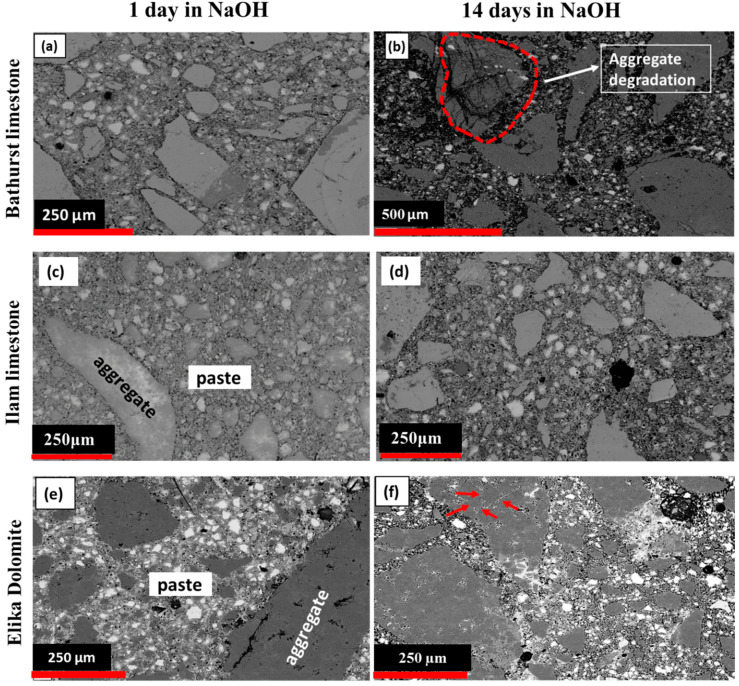
Comparison of SEM micrographs of mortar bars: Bathurst limestone (**a**) (1 day), (**b**) (14 days), Ilam limestone (**c**) (1 day), (**d**) (14 days), dolomite (**e**) (1 day), and (**f**) (14 days) immersed in NaOH solution.

**Figure 8 materials-15-04289-f008:**
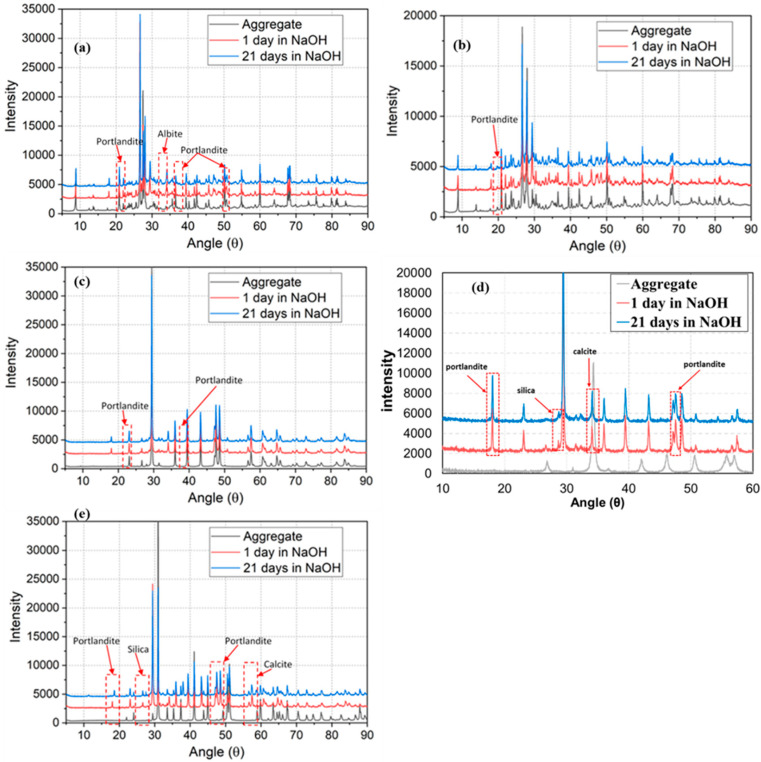
XRPD intensity comparison between fresh aggregate and accelerated mortar sample for (**a**) granite, (**b**) rhyodacite, (**c**) Bathurst limestone, (**d**) Ilam limestone, and (**e**) dolomite.

**Figure 9 materials-15-04289-f009:**
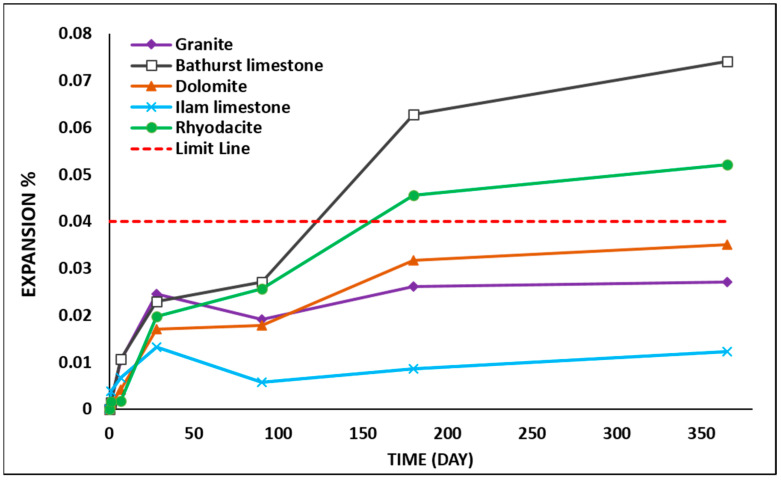
Concrete prism test results (ASTM C 1293).

**Figure 10 materials-15-04289-f010:**
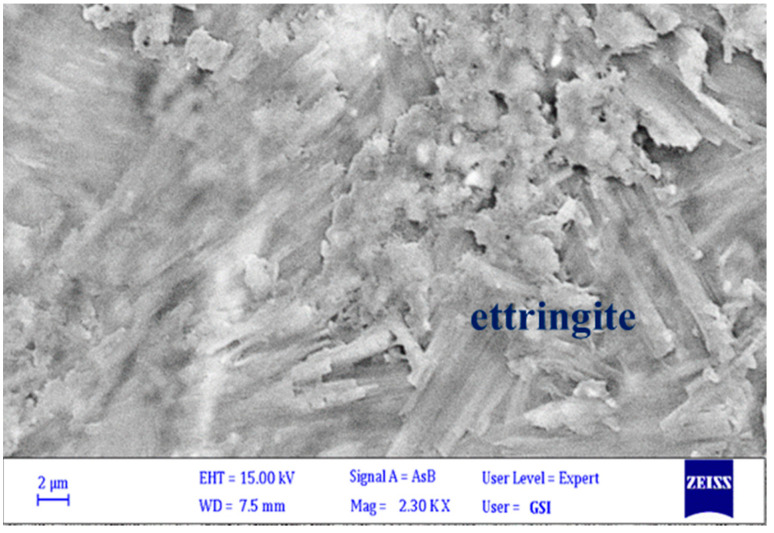
SEM micrographs of ettringite in the Bathurst limestone concrete prism samples after 6 months.

**Figure 11 materials-15-04289-f011:**
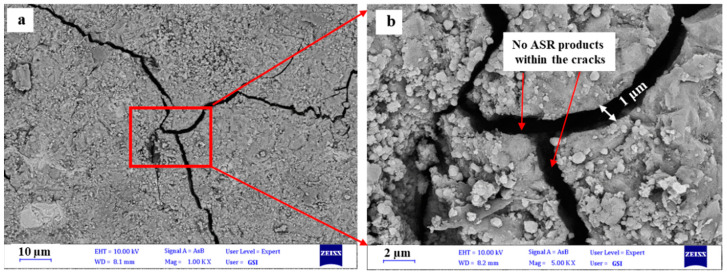
BSEM micrographs of the dolomite aggregate concrete prism samples after 6 months: (**a**) the micrograph with 10 µm magnification; (**b**) the micrograph with 2 µm magnification.

**Figure 12 materials-15-04289-f012:**
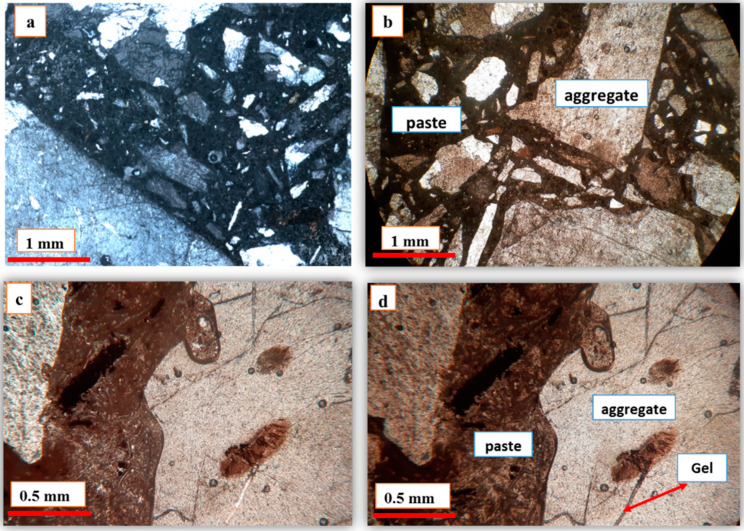
Optical microscopy photograph: (**a**) granite concrete after 1 month; (**b**) granite after 6 months; (**c**) rhyodacite concrete after 6 months with non-polarized light; (**d**) rhyodacite aggregate after 6 months (gel-like materials are seen in the polarized light).

**Figure 13 materials-15-04289-f013:**
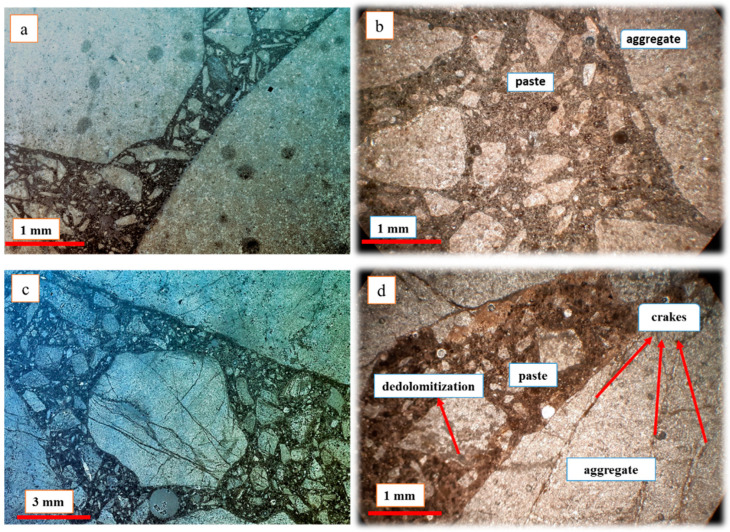
Optical microscopy photograph: (**a**) Ilam limestone concrete after 1 month; (**b**) Ilam limestone after 6 months; (**c**) dolomite concrete after 1 month; (**d**) dolomite aggregate after 6 months (aggregate cracking and dedolomitization are seen in the picture).

**Table 1 materials-15-04289-t001:** X-ray diffraction results of different aggregates.

Aggregate	Main Compositions	Trace Elements
Granite	Quartz, albite, orthoclase, biotite	Amphibole, calcite, pyrite
Rhyodacite	Quartz, albite, orthoclase, biotite	Amphibole, sanidine, sericite, calcite, hematite, kaolinite
Bathurst limestone	Calcite, quartz	
Ilam limestone	Calcite	Quartz
Dolomite	Dolomite, calcite	Quartz

**Table 2 materials-15-04289-t002:** X-ray fluorescence analysis of the aggregates.

Major Oxides		Plutonite	Volcanite	Sediment (Limestone)		
Analyte	LLD	Granite	Rhyodacite	Bathurst	Ilam	Dolomite
SiO_2_	0.03	71.96	70.25	4.84	1.54	0.15
Al_2_O_3_	0.01	13.82	14.74	1.25	0.48	0.03
Fe_2_O_3_	0.01	2.65	2.76	0.80	0.31	0.59
CaO	0.01	1.72	1.98	51.35	54.23	31.19
MgO	0.01	0.43	0.67	0.64	0.36	20.80
P_2_O_5_	0.01	0.09	0.13	0.10	0.05	0.04
TiO_2_	0.01	0.22	0.29	0.06	0.00	0.00
Na_2_O	0.01	2.85	3.95	0.22	0.14	0.03
K_2_O	0.01	5.30	4.25	0.10	0.07	0.01
MnO	0.01	0.05	0.06	0.18	0.11	0.03
CO_2_	-	0.91	0.92	40.46	42.71	47.13
Analytical SUM		100.00	100.00	100.00	100.00	100.00

**Table 3 materials-15-04289-t003:** Physical and mechanical properties of the aggregates.

Name	γ_d_ (KN/m^3^)	γ_sat_ (KN/m^3^)	n_e_ (%)	W_a_ (%)	V_p_ (m/s)	V_s_ (m/s)	UCS (MPa)
Granite	28.02	28.10	0.7	0.27	5126	2654	112.6
Rhyodacite	25.32	25.68	3.65	1.42	5067	2409	96
Bathurst limestone	26.22	26.56	2.22	1.45	4679	2423	48.78
Ilam limestone	26.66	26.68	1.97	0.84	4976	2657	58.56
Dolomite	27.31	27.59	1.04	0.78	4823	2575	61.92

Notes: γ_d_, dry unit weight; γ_sat_, saturated unit weight; n_e_, effective porosity; W_a_, water absorption by weight; G_S,_ specific gravity; V_p_, P-wave velocity; V_s_, S-wave velocity; UCS, uniaxial compressive strength.

## Data Availability

The data presented in this study are available upon request from the corresponding authors.

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
