# Peer review of "Assessment of Alkali–Silica Reaction Potential in Aggregates from Iran and Australia Using Thin-Section Petrography and Expansion Testing"

_materials, 2022, doi:10.3390/ma15124289_

Round 1
Reviewer 1 Report
This manuscript was well written titled as “Assessment of alkali-silica reaction-potential in aggregates from Iran and Australia using thin section petrography and expansion testing”. There are some questions about the contents, please carefully consider. Thanks.
1. The used cement or concrete should be introduced to show the alkali surroundings, due to that different cement may cause the various alkali-silica reaction, as well as the components of aggregate.
2. CO2 of sediment in Table 2 was up to 47%, but γd of sediment in Table 3 was similar as other materials, it needs some explanations.
3. In lines 313 to 319, the cracks were clear in one aggregate from Fig.7(b), while the differences of the other aggregate were not obvious from Fig.7(a) and 7(b), please carefully verified.
4. Some explanations of the two figures in Fig.11 missed, do they have differences?
5. As for the XRD tests, the intensity was compared between 1-day and 21-day, while the other tests was 1 month and 6 months, how does the rationality considered between the results from different tests?
6. In Fig.13, the scaleplate in (a), (b) and (d) were 1mm, while it was 3mm in (c), what does this intent to do? Some other places, such as in Fig.4, 0.5mm and 1mm?
7. All the five materials in Table 1 have quartz, how about calcite?
8. The references were too many and too general, it was necessary to refine and check for the special points.
Reviewer 2 Report
I have to admit that i had to read the entire paper at least three times in order to spot something wrong and did not spot any trouble. The introduction and the investigations are exposed in a very excellent way and there is not a single point in which i had to check again what i was reading. It rarely happen.
Even though the final conclusions about the limestone could have been easly predicted it still represent a valid result that added together wth the other investigations are in my humble opinion of high interest to readers. The english too is clear and never ambiguos. Sadly for me, the reviewer, i have nothing to suggest to the authors but the need of a more long-term follow up of the materials under investigation since the release of alkali can typically require a lot of time more.
The paper is ready to be accepted in the present way.
Reviewer 3 Report
The alkali-silica reaction-potential in aggregates was studied in this manuscript, with the help of thin section petrography and expansion testing. Its research program and finding have been presented clearly. It can be accepted after minor revision. Here are some comments for improvement: 1) the scale in picture e of 0.5 cm, presented in Figure 4, could be wrong? Please double check it. 2) the fourth picture in Figure 6 should be labelled as (d), instead of (b). 3) The authors are mis and mix-using the XRPD and XRD. Please revise it. 4) the x axis label of (a) and (c) in Figure 8 is missing. 5) What’s the crack width for the cracking in dolomite specimens. Please provide this information in Figure 11. The authors provided list of chemical evaluation and mechanic test, with detailed research data, which makes the manuscript looks quite well. But they should pay more attention to their writing. Too much writing errors exit in this manuscript.
Round 2
Reviewer 1 Report
It presents well. I agree for the publication.